# Emerging Nano- and Micro-Technologies Used in the Treatment of Type-1 Diabetes

**DOI:** 10.3390/nano10040789

**Published:** 2020-04-20

**Authors:** Rosita Primavera, Bhavesh D Kevadiya, Ganesh Swaminathan, Rudilyn Joyce Wilson, Angelo De Pascale, Paolo Decuzzi, Avnesh S Thakor

**Affiliations:** 1Interventional Regenerative Medicine and Imaging Laboratory, Department of Radiology, Stanford University, Palo Alto, CA 94304, USA; rprima@stanford.edu (R.P.); kevadiya@stanford.edu (B.D.K.); gswami@stanford.edu (G.S.); joycewilson@stanford.edu (R.J.W.); 2Laboratory of Nanotechnology for Precision Medicine, Fondazione Istituto Italiano di Tecnologia, 16163 Genoa, Italy; 3Unit of Endocrinology, Department of Internal Medicine & Medical Specialist (DIMI), University of Genoa, 16163 Genoa, Italy; angelo.depascale@hsanmartino.it

**Keywords:** diabetes, glucose sensor, nanoparticles, microparticles, tissue engineering

## Abstract

Type-1 diabetes is characterized by high blood glucose levels due to a failure of insulin secretion from beta cells within pancreatic islets. Current treatment strategies consist of multiple, daily injections of insulin or transplantation of either the whole pancreas or isolated pancreatic islets. While there are different forms of insulin with tunable pharmacokinetics (fast, intermediate, and long-acting), improper dosing continues to be a major limitation often leading to complications resulting from hyper- or hypo-glycemia. Glucose-responsive insulin delivery systems, consisting of a glucose sensor connected to an insulin infusion pump, have improved dosing but they still suffer from inaccurate feedback, biofouling and poor patient compliance. Islet transplantation is a promising strategy but requires multiple donors per patient and post-transplantation islet survival is impaired by inflammation and suboptimal revascularization. This review discusses how nano- and micro-technologies, as well as tissue engineering approaches, can overcome many of these challenges and help contribute to an artificial pancreas-like system.

## 1. Introduction

### 1.1. Diabetes Mellitus

Diabetes mellitus (DM) is a chronic metabolic disorder characterized by elevated blood glucose levels (BGLs). It is one of the most challenging global health issues affecting more than 450 million people worldwide and this number is estimated to increase to 693 million by 2045 (Figure 1) [1,2,3]. In the US, the cost of diabetes management exceeds $322 billion every year with the increasing emergency room visits due to diabetes-related complications. 

Currently, diabetes is diagnosed using one of the following criteria: glycated hemoglobin (HbA1c) ≥ 6.5% (48 mmol mol^−1^); fasting plasma glucose ≥ 126 mg dL^−1^ (7.0 mmol L^−1^), in which fasting is defined as no caloric intake for at least 8 h; two-hour plasma glucose ≥ 200 mg dL^−1^ (11.1 mmol L^−1^) during an oral glucose tolerance test (OGTT); or random plasma glucose ≥ 200 mg dL^−1^ (11.1 mmol L^−1^) [4]. Based on the pathogenesis, the American Diabetes Association has classified diabetes into four categories: (a) Type 1 DM (T1DM), also known as insulin-dependent or juvenile diabetes, occurs in only 5–10% of all diabetes cases, results from a cellular-mediated autoimmune destruction of the pancreatic beta cells (*β*-cells); (b) Type 2 DM (T2DM), accounts for 90–95% of all diabetes cases, with an etiology that is not very clear but is characterized by insulin resistance with relative insulin deficiency, is frequently due to progressive loss of pancreatic *β*-cell secretion [4]; (c) Gestational DM, defined as any degree of glucose intolerance with onset or first recognition during the second or third trimester of pregnancy, is associated with adverse pregnancy outcomes including fetal macrosomia, stillbirth, neonatal metabolic disturbances and related problems [4,5]; (d) Specific types of DM are defined by diabetes with unusual causes such as monogenetic defects in *β*-cell function (e.g., neonatal diabetes and maturity-onset diabetes of the young (MODY)), genetic abnormalities of insulin action, diseases of the exocrine pancreas (e.g., cystic fibrosis and pancreatitis), endocrinopathies (e.g., agromegalia), drug- or chemical-induced diabetes (e.g., glucocorticoids), and infections (e.g., congenital rubella, cytomegalovirus). 

The main approach in treating and managing diabetes consists of multiple daily injections of insulin together with the continuous and accurate monitoring of BGLs in order to maintain their values within the normoglycemic range of 70–140 mg dL^−1^. Unfortunately, insulin self-administration is associated with pain and often inadequate glucose control, resulting in acute and chronic complications. These complications include (i) microvascular diseases such as diabetic retinopathy, nephropathy and neuropathy, and (ii) macrovascular diseases such as heart attacks, strokes, and insufficient blood flow to a leg. 

However, in the last few decades micro-, nano-technologies and cell-therapies have provided several solutions to potentially overcome the limitations of conventional insulin therapy, thereby providing patients with improved disease management and quality of life. 

### 1.2. Insulin

Exogenous insulin therapy is the primary approach for treating diabetes, especially T1DM, and is also sometimes required in cases of advanced T2DM. Insulin is a dipeptide hormone secreted by the *β*-cells, which are located within pancreatic islets. It consists of 51 amino acids with a molecular weight of 5802 Da and is made of two chains (A and B) inter-linked by disulfide bridges. The A chain comprises 21 amino acids with an N-terminal helix linked to an anti-parallel C-terminal helix, while the B chain has 30 amino acids with a central helical segment. The two chains are connected by two disulfide bonds through the N- and C-terminal helices of the A chain and the central helix of the B chain [6,7].

Insulin is a key molecule that regulates the intracellular transport of glucose into insulin-responsive tissues, such as skeletal muscle, liver and adipose tissue. Upon its release, insulin binds to specific membrane-bound insulin receptors on the target cells, which then results in the activation of metabolic processes within these cells. The insulin receptor is a heterotetrameric structure consisting of 2*α* and 2*β* glycoprotein subunits linked by disulfide bonds. The first step in activation of the insulin signaling pathway is the binding of insulin to the extracellular *α* subunit, which results in a conformational change to allow ATP-mediated phosphorylation of the intracellular *β* subunit, followed by tyrosine phosphorylation of the intracellular substrate proteins known as insulin-responsive substrates (IRS). The second step involves the activation of insulin signaling pathway by IRS, resulting in the recruitment and translocation of the glucose transporter type 4 (GLUT4) to the cell membrane (Figure 2) [8].

Insulin was discovered by Sir Frederick G. Banting in 1921 and it was subsequently purified by James B. Collip. The first forms of insulin used for clinical applications were extracted from porcine or bovine pancreas. Although their actions are very similar to human insulin, bovine and porcine insulin show higher immunogenicity than human insulin and are chemically different in three (in bovine insulin) and one (in porcine insulin) amino acid residues. Thus, the current human-sequence insulin is generated either by recombinant DNA technology or by enzymatic modification of porcine insulin. Since the half-life of insulin in plasma is very short (4–6 min) due to the rapid uptake and degradation of this molecule in the kidneys and liver, efforts have been taken to develop commercial insulin variants with tunable pharmacokinetics [8]. Variants with a rapid and/or prolonged activity have been obtained by formulating insulin either as a soluble preparation (regular or neutral insulin) or as a complex with protamine and/or zinc (see Table 1) [9,10,11]: (a) protamine can be used to create an intermediate-acting complex with isophane (isophane insulin—also known as NPH (neutral protamine Hagedorn) insulin) and is obtained by mixing neutral insulin with isophane at different ratios, the most common is composed of 30% neutral insulin and 70% isophane; (b) zinc can be used to stabilize the hexamer aggregate of insulin molecules and give an intermediate-acting complex; (c) protamine and zinc can both be used to create a long-acting protamine-zinc insulin (though, this is rarely used because it is available only as a bovine insulin).

Conventional administration of insulin is almost entirely performed with subcutaneous injections at different sites (i.e., upper arms, thighs, buttocks and abdomen) using syringes, pens or intravenous infusion for emergencies. Subcutaneous administration is often associated with pain and poor patient compliance due to needle phobia, skin bulges, allergic reactions, common infections and stress generated from the difficult long-term regimen of insulin therapy. 

Currently, insulin replacement therapy is adopted to mimic natural fluctuations in insulin levels throughout the day. The typical treatment consists of multiple pre-prandial injections of rapid-acting insulin at mealtimes and pre-dormital long-acting insulin to provide a basal level of insulin [12,13,14]. The disadvantage of this method is that even with frequent self-monitoring of blood glucose level and administration of insulin, conventional therapy produces far from ideal physiological response. In addition to the pain, the gap between glucose monitoring and insulin dosage, combined with the delayed absorption of subcutaneously injected insulin, limits blood glucose control and can lead to hypoglycemia [15]. Inadequate glucose control can result in fluctuations in BGLs which, if chronic, can increase the risk of developing cardiovascular disease, stroke, non-healing wounds, blindness, cancer and many other co-morbidities. Acute complications can occur from hypoglycemia, which if severe, can result in coma, brain damage or even death [12,13]. 

To improve the quality of life for diabetic patients, various alternative methods have been explored to achieve a controlled long-term release of insulin and non-invasive administration (i.e., oral, nasal, pulmonary or transdermal). Although different preclinical and clinical trials are undergoing for oral (ClinicalTrials.gov NCT00982254, NCT03430856, NCT01035801, NCT00668850), transdermal (ClinicalTrials.gov NCT00519623), and nasal (ClinicalTrials.gov NCT02485327) insulin delivery, realizing an “ideal pancreas-like insulin delivery system” remains challenging, because of the poor shelf-life of insulin due to its fast chemical and enzymatic degradation, and the poor permeability of physiological barriers (i.e., gastrointestinal, nasal, pulmonary, transdermal barriers). 

## 2. Platforms for Carrying Insulin 

Several technologies (i.e., nano-, micro-technology) have been developed to overcome the drawbacks of insulin replacement therapy and improve patient compliance. Using platforms or devices at the nano- or micro-scale through non-invasive routes (i.e., oral, nasal, pulmonary, transdermal) has the potential to overcome the limitations of the conventional subcutaneous insulin injection and to realize a more patient-friendlier device, in order to improve patient experience and therefore compliance, greatly improving the quality of life for diabetic patients. 

By encapsulating insulin into specific organic or inorganic nano- or micro-delivery carriers (Figure 3), the pharmacological properties (e.g., solubility, circulation half-life, and toxicity) of insulin can be improved dramatically, thereby leading to safer and more efficient treatments [14,15]. Such carriers can be made to (i) enhance therapeutic efficacy and minimize adverse effects, (ii) prevent premature degradation and denaturation after interaction with the physiological environment, (iii) enhance uptake through physiological barriers, such as the skin and intestinal wall mucosa, and/or (iv) incorporate stimuli-responsive elements for on-demand release.

Nanoparticles were first developed around 1970 as carriers for ophthalmic and oral delivery of drugs. These particles are submicronic (<1 µm) and they include nanospheres and nanocapsules, based on the process used for their preparation. Nanocapsules are vesicular systems in which the drug is loaded into a central cavity and is surrounded by a unique polymeric membrane; nanospheres are matrix systems containing a drug dispersed throughout the particle. In contrast, microparticles consist of polymeric particles with sizes ranging from 50 nm to 2 mm. Depending on the process used for the preparation of microparticles, microspheres or microcapsules can be obtained. Microspheres and microcapsules are the morphological equivalents of nanospheres and nanocapsules, respectively.

Numerous fabrication processes can be used for the preparation of nano- and microparticles including solvent evaporation, organic phase separation, interfacial polymerization, emulsion polymerization, and spray drying, although only a few are acceptable for insulin formulation particles. The physicochemical properties of insulin make their formulation challenging as inactivation is possible during their incorporation into nano- or microparticles. The difference in size between nano- and microparticles have numerous effects. Water penetrates nanoparticles more rapidly, resulting in an increased burst release and generally more rapid release kinetics. Furthermore, accelerated drug release due to particle deterioration is noted when polymers are degraded by hydrolysis. Nanoparticles are more likely to aggregate, but the surfaces of particles can be functionalized to optimize binding to specific receptors (Table 2). A variety of materials are currently used for preparing insulin-loaded nano- and microparticles, including chitosan, alginate, phospholipids, and synthetic polymers, like poly(lactic acid-glycolic acid) (PLGA) [16,17]. 

### 2.1. Nano-Scale Carriers

Insulin-loaded nanoparticles have recently attracted considerable interest because they can enhance the stability of insulin by preventing its enzymatic and hydrolytic degradation. For example, Lin et al. showed that chitosan-insulin nanoparticles prolonged the residence time of insulin in the small intestine and enhanced the permeation of insulin into the bloodstream via paracellular pathways. Chitosan is a non-toxic cationic polysaccharide, which has been used as a permeation enhancer for the absorption of insulin and other hydrophilic molecules [18]. Chitosan adheres to the mucosal surface and opens the tight junctions (TJs) between epithelial cells. Chitosan particles infiltrate the mucus layer and transiently open the TJs where they showed a sustained effect in decreasing BGLs over at least 10h in diabetic rats [19]. To avoid degradation of insulin-nanoparticles in the gastric environment, Cui et al. synthesized pH-sensitive carboxylated chitosan-grafted poly(methylmethacrylate) nanoparticles—these nanoparticles exhibit quick insulin release in the neutral environment (pH = 6.8–7.4) of the small intestine, but not under acidic conditions (pH = 2.0) of the stomach [20]. Similarly, chitosan-coated liposomes, which consist of nanoparticles made by phospholipids, presented a higher mucoadhesion capability and a greater hypoglycemic effect [21,22]. Insulin loaded in PLGA nanoparticle carriers are able to protect insulin from degradation in the gut, thereby leading to better absorption across the intestinal epithelium, especially when targeting molecules are applied on the particle surface [15,23]. PLGA is a Food and Drug Administration (FDA)-approved biodegradable and biocompatible synthetic polymer, which degrades by hydrolysis of its ester linkages in the presence of water into two monomers, lactic acid and glycolic acid. It is relatively easy to functionalize. For instance, insulin-loaded PLGA nanoparticles decorated with Fc fragments are efficiently transported across the intestinal epithelium and the absorption efficiency is 10× higher compared to non-targeted particles [24]. Based on a similar approach, Jain et al. showed higher insulin uptake of L-Valine-conjugated PLGA nanoparticles (approximately 91%) as compared to unconjugated particles (approximately 48%), due to their relatively high affinity for oligopeptide transporters in the intestine, increasing bioavailability and improving therapeutic responses for orally-administered insulin [25]. In another example, Zhang et al. demonstrated that chitosan coating makes PLGA particles more mucoadhesive compared to uncoated ones, thereby prolonging insulin retention on the mucosa resulting in improved oral bioavailability [26]. Other research groups have studied the efficacy of pH-sensitive particles composed of PLGA and pH-sensitive polymers (e.g., hydroxypropyl methylcellulose phthalate (HP55)); following oral administration of these particles, less than 15% of insulin is released in the stomach (pH = 1.2), while in the intestine (pH = 7.4), about 90% insulin is released from nanoparticles [27].

Another nanocarrier explored for insulin delivery consists of biodegradable bilayer vesicles made out of amphiphilic phospholipids, commonly called liposomes. Insulin encapsulated in liposomes is sufficiently protected from pH, enzymatic degradation and immune recognition. For example, Kisel et al. demonstrated stabilization of BGLs in rats using insulin-loaded liposomes [28]. Dwivedi et al. further demonstrated that a silica coat can enhance the stability of phosphatidylcholine liposomes resulting in being more efficacious in reducing glucose levels in rats compared to uncoated vesicles [29]. Modifications of the liposomal surface has also been investigated to improve the stability of liposomes in the gastrointestinal environment and promote their adhesion to the intestinal epithelium. For example, liposomes coated with poly (ethylene glycol) (PEG) 2000, or sugar chains of mucin, showed an improved oral insulin delivery compared to uncoated liposomes [30]. 

Diasome Pharmaceuticals (Conshohocken, PA, USA) is developing a liposome-based drug delivery system, called hepatic-directed vesicles—insulin (HDV-I), for the subcutaneous or oral administration of insulin. Data from a placebo-controlled, dose-ranging trial for oral HDV-I demonstrated in six patients with T2DM a statistically significant reduction in blood glucose levels over a period of 14 h as compared to placebo [31]. Diasome has recently launched an 18-weeks phase II/III proof-of-concept study (ClinicalTrials.gov NCT00814294) in metformin failing patients with T2DM for studying HbA1c drops. 

Transferosomes, which are elastic/deformable lipid vesicles, have been used for transporting insulin across the skin with a relatively high degree of efficiency. Transferosomal insulin (Transfersulin^®^) that is applied to intact human skin can reduce plasma glucose concentration by ∼20% within 3–4 h. Several clinical studies have shown that Transfersulin^®^ allows insulin to reach the systemic circulation and maintain normoglycemia over several hours [32].

### 2.2. Micro-Scale Carriers

Microparticles allow a slower and sustained release of insulin because of their large size. In contrast to nanoparticles, microparticles injected into a variety of tissues tend to stay where they are placed. For example, Kohane et al. demonstrated increased retention of large polymeric particles injected at the sciatic nerve for more than 8 weeks [33,34]. Similarly, at least two weeks of retention was observed when microparticles that are 5, 25, 60, and 250 µm in diameter were injected into the peritoneum of mice when compared to an equal mass of nanoparticles, which showed complete clearance in the same time frame [33,35].

PLGA microspheres subcutaneously injected can produce a sustained release of insulin for several days because of their larger size compared to nanoparticles. In particular, Hinds et al. showed that PEGylated human insulin encapsulated in PLGA microspheres produced <1% release of insulin immediately after a single subcutaneous injection, thereby avoiding toxic effects of over-administration of insulin and lowering the BGLs of diabetic rats to values <200 mg dL^−1^ for approximately 9 days [36]. Recently, Primavera et al. proposed a hierarchical system, in which nanoscale granules of insulin were synthesized and encapsulated within the porous matrix of PLGA microplates which were square in shape. Insulin granules-loaded microplates showed a very slow release profile of insulin over time and lowered BGLs in vivo as compared to untreated mice [37]. Cheng et al. studied the bioavailability of magnetite nanocrystals and insulin co-encapsulated into PLGA microparticles, evaluating their effects on hypoglycemia in mice in the presence of a circumferentially applied external magnetic field. Insulin-magnetite-PLGA microparticles orally administered in the presence of an external magnetic field showed a substantially improved hypoglycemic effect in mice, suggesting that magnetic force can be used to improve the efficiency of orally delivered insulin [38].

Insulin-loaded alginate-chitosan microspheres that were orally administered were shown to efficiently maintain lower BGLs for longer periods of time (approximately 60 h) [39,40]. Other natural and mucoadhesive polymers such as tragacanth or gelatin have also been used to deliver insulin. Nur and Vasiljevic prepared insulin particles by the inclusion of the drug into a tragacanth hydrogel to protect the peptide from the strong acidity of the gastrointestinal environment [41]. Wang et al. demonstrated that aminated gelatin, a positively charged gelatin derivative, can significantly increase nasal absorption of insulin due to its mucoadhesive properties and positive charge. The enhanced absorption of insulin was attributed to the increased paracellular permeability via opening of the tight junctions as a result of the temporary dehydration of the nasal epithelial mucosa [42]. Furthermore, Emisphere’s oral insulin (based on Eligen^TM^ Technology, Roseland, NJ, USA) uses a permeation enhancer to favor the oral absorption of insulin. Emisphere reported completion of a placebo-controlled four treatment arm, 90 days phase II study, showing that Eligen^TM^ improved both glycemic control and insulin sensitivity while demonstrating a good safety profile with no significant differences in hypoglycemic events, serious adverse events or insulin antibody formation in comparison to placebo [43]. Nobex Inc. (Research Triangle Park, NC, USA) combined a modified insulin (hexyl PEG insulin-monoconjugate 2 (HIM2)) with a penetration enhancer and sodium cholate. A study conducted in healthy volunteers indicated that the orally administered HIM2 suppressed endogenous glucose production faster and for longer as compared to injectable insulin [44]. Biocon is developing the next HIM2 generation, called IN-105 (insulin modified with a small PEG). In early clinical studies, IN-105 was tested on 20 patients with T2DM poorly responding to metformin. IN-105 reduced the 2 h postprandial glucose excursion in a dose-dependent manner. It was well tolerated by all the patients and caused symptomatic, biochemically-confirmed hypoglycemia in only one patient at 20 mg dose [45]. Biocon has initiated a longer-term study (Trial ID: CTRI/2009/091/000479) with IN-105 in patients with T2DM. 

Microparticles are also frequently used in pulmonary delivery because they preferentially deposit in the deep lungs and do not aggregate under shear forces. Advanced Inhalation Research, a company that is now part of Alkermes/Eli Lilly, has produced large porous insulin particles of low-mass and density with an aerodynamic-diameter of 1–3 μm. These particles are produced by encapsulating insulin into a biodegradable polymer matrix. They have a reduced tendency to aggregate that facilitates dispersion and diminishes susceptibility to local phagocytosis. Technosphere^®^ Insulin (Afrezza) is a novel inhalation powder based on microparticles formed by the self-assembly of crystals of fumaryl diketopiperazine and presenting an average diameter of 2–2.5 μm [46]. The formulation self-assembles into an ordered lattice array at low pH that then dissolves at the neutral pH of the alveolar surface to release the insulin rapidly. The formulation is designed for use with the MedTone^®^ dry-powder-inhaler delivery device, which is a capsule-based, high-impedance inhaler that uses a passive powder-deagglomeration mechanism. Afrezza was FDA approved in 2014 and various clinical trial are currently being performed (ClinicalTrilas.gov NCT02485327, NCT03143816, NCT01874392, NCT00662857, NCT01544881) [47]. In 2010, Hamishenkar et al. demonstrated that insulin-loaded PLGA microcapsules showed a sustained release profile resulting in a longer mean residence time compared to the control treatments, i.e., respirable spray-dried insulin powder administered by intratracheal insufflation and NPH (long-acting) insulin administered by subcutaneous (SC) administration (4 and 5 folds, respectively) [48]. Edward et al. showed that not only the size of particles but also their porosity affect insulin bioavailability. They demonstrated that inhaled large porous insulin-loaded PLGA particles resulted in elevated systemic levels of insulin and prolonged systemic glucose levels (96 h) compared to small nanoporous ones (4 h) [49]. In this direction, Lee et al. reported the application of bovine serum albumin as an osmotic agent for controlling the synthesis of porous PLGA microspheres. This strategy allowed the initial burst release of insulin entrapped in the porous structure of polymer microspheres to be reduced [50]. 

Another approach to avoid rapid drug absorption, degradation and to evade pulmonary clearance, while simultaneously prolonging inhalable insulin’s half-life at the lung, requires the use of inhaled formulations of insulin with stealth characteristics, preventing immune recognition. Hyaluronic acid (HA) is a naturally occurring polymeric hydrogel based on a linear polysaccharide, which is comprised of repeating units of D-glucuronic acid and N-acetyl-Dglucosamine, and linked by *β*-1,4 and *β*-1,3 glycosidic bonds. HA is naturally present in the lung, where its role includes the protection of lung elastin from the damage associated with inflammatory lung disease and in the repair of lung injuries [51]. For this reason, HA has been widely studied for pulmonary and lung applications [52]. For example, Surendrakumar et al. showed that HA spray-dried with insulin from aqueous solutions to form inhalable dry powders increased the mean residence time and the terminal half-life of insulin to the lungs of male Beagle dogs compared to spray-dried pure insulin [53].

## 3. Platforms for Transdermal Delivery of Insulin

Microneedles (MNs) are micron-scale devices capable of circumventing subcutaneous barriers and improving penetration through the skin to facilitate transdermal insulin delivery. Microneedles are able to puncture the skin and create micrometer channels that transiently open the stratum corneum, thereby penetrating across the epidermis. Generally, microneedles are <300 µm in diameter, with a needle-shaped device which is 50–900 µm long, which can deliver hydrophilic small drugs, peptides, proteins and nanoparticles [54]. The first one described for drug delivery was introduced in 1979 by Gerstel and Place [55].

As reported by Bariya et al., these devices can be classified into four categories by the different approaches used for passage through the skin: (a) a “poke and patch” approach that involves application of a medicated patch to skin after microneedle treatment; (b) a “coat and poke” method that involves coating of a solid microneedle with a uniform layer of a drug containing solution (used for skin administration of unstable drugs with precise dosages); (c) a “poke and release” approach that utilizes drug-encapsulated solid MNs, which consists of dissolving/biodegradable polymers to have in one-step both skin penetration and drug release; and (d) a “poke and flow” approach that consists in a hollow MN which facilitates the direct infusion of liquid insulin preparations into the skin without removal of the MNs [56]. 

Compared to conventional insulin administration, microneedles have many benefits which include the fact that they are painless, avoid the first-pass metabolism and show more rapid injection site healing than conventional hypodermic needles [54,56]. Following the “poke and patch” approach, Martanto et al. placed an insulin solution on the surface of an array containing 105 mm × 1 mm long, solid microneedles and utilized a high-velocity injector to implant these arrays within the skin of diabetic hairless rats; they demonstrated an increase in transdermal delivery of insulin and a rapid and steady reduction in BGLs (up to 80%) [57]. 

Migalska et al. adopted the “poke and release” strategy, revealing that insulin-loaded MNs, obtained from aqueous blends of 20% *w*/*w* poly (methylvinyl)ether maleic anhydride with laser-engineered molds, significantly enhanced insulin transport across neonatal porcine skin [58]. Similarly, Liu et al. proposed microneedle made out by hyaluronic acid (HA-MNs) as an effective and safe method for transdermal insulin delivery into the systemic circulation without inducing serious skin damage. Also, pharmacodynamics and pharmacokinetic analyses showed that HA-MNs were almost completely absorbed from the skin to reach the systemic circulation, and the hypoglycemic effect was comparable to that of the subcutaneous injection of insulin [59]. Similar to the above study, transdermal delivery of insulin-loaded dissolving MN patches, composed of starch and gelatin, was found to be completely dissolved within 5 min after insertion into the skin and resulted in a rapid and efficient absorption of insulin [60]. Recently, Kim et al. developed a novel dissolving microneedle fabrication technique called the droplet-born air blowing, in which the polymer droplet is shaped to the microneedle via air-blowing [61].

Nordquist et al. studied the “poke and flow” approach to deliver insulin lispro in diabetic rats. Their study showed that an array of hollow 21 µm × 400 µm long microneedles, connected to an electrically controlled drug dispenser with a 12 µL reservoir, increased the plasma insulin level and minimized risks for diabetes-induced complications [62].

PassPort^TM^ system for the delivery of insulin through transdermal patches was one of the first insulin skin patches made by Altea Therapeutics, US, in 2007. The insulin skin patch system is designed to improve and sustain levels of basal insulin to lower the risk of hypoglycemia, as well as ensure greater patient compliance than injecting insulin equivalents. It also has the advantage of being stored at room temperature (ClinicalTrials.gov NCT00519623). In a recent clinical study, Norman and collaborators tested in 16 children and adolescents with T1DM the efficacy of a single hollow-microneedle device to deliver insulin lispro and compared it with a subcutaneous insulin pump catheter. Microneedle administration of insulin resulted in less needle insertion pain and faster insulin onset and offset (ClinicalTrials.gov NCT00837512) [63]. Despite these benefits, insulin-based microneedle still presents several drawbacks, including less precise dosage accuracy than hypodermic needles due to differences in individual skin layer thickness and skin hydration, drug delivery problems associated with non-vertical application of the microneedle to the skin, damage to veins after repeated use, and potential breakage of the microneedle tip within the skin (Table 2) [54,56].

## 4. Platforms That Can Modulate Insulin Delivery 

Nano- and micro-particulate platforms have shown good potential to minimize the frequency of insulin injections and overcome the limitations related to the harsh environment of the gastrointestinal tract (acidic pH), the variable and unpredictable inhalation efficiency through the nasal and pulmonary routes, and limited transport across epithelial barriers [64]. However, the dosage of insulin administered for the aforementioned alternative routes is much higher than the traditional dosage used subcutaneously and it could potentially increase the risk of initiation of some disorders [64,65,66]. Since insulin is a potent mitogen, it can induce effects of a growth factor both at cellular and systemic levels due to its affinity to bind to the type-1 insulin-like growth factor receptor (IGF). Therefore, mimicking the dynamic release of insulin is critical to avoid extended tissue exposure to hyperglycemia or the adverse risk of hypoglycemia.

Since 1979 studies have investigated insulin formulations with a glucose-responsive activity that allow insulin delivery at a precise dosage [67]. Accordingly, an automated pump containing insulin and glucagon (a hormone that has the opposite effect of insulin) has been studied recently among both adults and adolescents with T1DM. This dual delivery system improves the glycemic control while reducing the frequency of hypoglycemic episodes (ClinicalTrials.gov NCT01762059, NCT01833988) [68]. Despite these advantages, automated pumps are expensive, carry risks associated with implanted sensors, and the use of a permanent cannula increases the patient’s risk of infection and inflammation. Thus, as an alternative, recent advancements in polymer engineering and nanotechnology have led to investigations on insulin-delivering particles that carried glucose-responsive molecules, which can accurately mimic the physiological response to blood glucose changes. Such platforms typically employ one of three main classes of molecules able to sense and immediately react to the presence of glucose: glucose oxidase (GOx); glucose-binding small molecules including boronic acid derivatives like phenylboronic acid (PBA); and glucose-binding protein like Concavalin A (ConA) (Figure 4, Table 3) [69]. These molecules can detect fluctuation in blood glucose levels and release insulin cargo in response through degradation, swelling or disassembly of the particulate formulation [70,71].

### 4.1. Glucose Oxidase-Based System

Glucose-oxidase is the most commonly used glucose sensor due to its high affinity for glucose [72]. It is a homodimer composed of two identical 80 kDa subunits, highly specific for glucose (Table 3). Glucose-oxidase catalyzes the oxidation of glucose to gluconolactone, which can then be hydrolyzed to gluconic acid to generate hydrogen peroxide in the presence of oxygen. Consequently, GOx is generally incorporated in pH-sensitive materials that are degraded with local acidification in response to elevated glucose levels thus triggering the release of insulin or other antidiabetic molecules. Based on this strategy, several polymeric hydrogel matrices or membranes that swell or shrink in response to variations in local glucose concentrations have been developed [73]. For example, pH-sensitive chitosan microgels containing large-pore silica particles loaded with GOx and cerium oxide nanoparticles (Ce-NPs, having an excellent antioxidant catalytic activity), can respond to hyperglycemia and self-regulate the release of insulin in vitro to mimic the repetitive change between hyper (400 mg mL^−1^) and normoglycemic (100 mg mL^−1^) state [74]. 

Gu et al. recently developed an injectable nano-network, consisting of oppositely charged dextran nanoparticles encapsulating insulin and GOx, which are capable of dissociating and releasing insulin in response to high BGLs. This nano-network showed a fast insulin release profile during hyperglycemic conditions and an insignificant release of insulin in normoglycemia. They also demonstrated that one single subcutaneous injection of the nano-network into T1DM mice was able to maintain BGLs within a normoglycemic range for up to 10 days [75]. The same group also developed an interesting platform by integrating H_2_O_2_-responsive polymeric vesicles, loaded with GOx and insulin, into a painless transcutaneous hyaluronic acid microneedle array patch. Both in vitro and in vivo studies showed that insulin release responded quickly to elevated glucose concentrations, and a single patch can modulate BGLs effectively avoiding the risk of hypoglycemia [76]. Using the same technology, they studied the efficacy of similar core-shell microneedle array patches made out of degradable cross-linked poly(vinyl alcohol) gel. The core of this device contains GOx that generates H_2_O_2_ to stimulate the release of insulin, while the shell is embedded with catalase, necessary to remove the excess of H_2_O_2_ produced by the GOx, which may be toxic in vivo [77]. In T1DM mice, they demonstrated that this smart patch, with a bioresponsive core and protective shell, could effectively regulate the BGLs within a normal range, thereby improving biocompatibility [76,78]. 

Recently, Volpatti et al. co-formulated acetylated dextran nanoparticles with GOx, catalase and insulin, and showed a sustained release of insulin with rapid release kinetics. Analyses of this particle in both healthy and diabetic (streptozotocin-treated) mouse models show direct evidence of glucose-responsiveness in vivo, rapid glycemic control, and prolonged normoglycemia for 16 h with a single subcutaneous injection in streptozotocin-treated mice [79]. While this observation is encouraging, the enzymatic conversion of glucose in gluconic acid is variable and slow due to the insufficient oxygen supply in vivo, which results in unpredictable insulin release from the delivery systems. In a confined subcutaneous tissue, the oxygen level is not proportional to glucose concentration, especially in the hyperglycemic state. Gu et al. attempted to solve this problem by incorporating catalase with glucose-oxidase, however catalase converted only half of the byproduct H_2_O_2_ in oxygen [75]. Alternatively, Li et al. employed MnO_2_ nanoparticles (instead of catalase) to continue to catalyze the conversion of H_2_O_2_ to molecular O_2_ and water at the one-to-one molar ratio. In this way MnO_2_ particles had a dual function: (i) preserving the enzymatic activity of glucose oxidase and (ii) avoiding the over-acidification at low glucose concentration [80,81]. 

### 4.2. Glucose-Binding Protein-Based System

Similar to glucose oxidase, glucose-binding proteins provide high specificity and binding to glucose (Table 3). In particular, lecithins are a group of carbohydrate-binding proteins, as natural receptor-based glucose-sensing elements. The most frequently used lecithin is plant lecithin purified from jack beans, concanavalin A (molecular weight 104,000 Da), which contain four identical binding sites for sugars, such as D-mannose and D-glucose [82]. Since the 80s, several seminal studies investigated the effectiveness of this compound due to its high affinity to D-glucose. Firstly, Brownlee et al. synthesized a stable, biologically active glycosylated insulin derivative complex with ConA. They showed that upon exposure to external glucose, insulin can be released via competitive binding between glucose and glycosylated insulin with ConA and the amount of hormone released has been shown to be proportional to the external glucose concentration [67]. Yin et al. developed glucose and pH dual-responsive microhydrogels based on ConA for self-regulated insulin delivery. The release studies revealed that in the presence of an increased glucose level, both the decrease in the crosslinking density and the increase in the hydrophilicity of the microhydrogels lead to the swelling of the hydrogels and a quick release of insulin [83]. Sato et al. prepared sugar-sensitive microcapsules spherical in shape with 3–10 µm diameter, using a layer-by-layer deposition of ConA/glycogen membrane on a calcium carbonate particle containing insulin. Insulin is released in response to high concentrations of D-glucose and other sugars because the added sugars replace glycogen in the binding site of ConA, resulting in the enhanced permeability of the microcapsules to insulin [84]. Nevertheless, ConA-based approaches suffer from possible toxicity, which could be avoided using a small dose of ConA (<10 µg µL^−1^) subcutaneously implanted [85]. Despite these concerns, there is still interest in developing Con-A based systems for clinical use. For example, SmartCells Inc. (part of Merck & Co., Inc., Kenilworth, NJ, USA) in 2010, developed a synthetic form of insulin-containing pendant sugar units that could be coupled with glucose-binding proteins for regulating on-demand insulin release (ClinicalTrial #MK-2640) [86]. That said, overcoming the host immune response still remains a challenge. 

### 4.3. Phenilboronic Acid-Based System

Small molecules like phenylboronic derivatives are Lewis acids known to bind reversibly to cis-1,2 or cis-1,3 diols, such as glucose, which stabilizes a negative charge on the boronic acid [87]. Natural glucose-sensing elements have some shortcomings which include high cost, instability, and antigenicity. While PBA can address these shortcomings, its lack of specificity for glucose because of their higher affinity for other diols in the body like fructose, and its pKa of around 9, hinders efficient binding to glucose at physiological pH (pH = 7.4) (Table 3). Alternatively, combining PBA with other monomers results in polymers with more suitable pKa values (from 7.4 to 9.0) and this makes them more capable of responding to glucose under physiological conditions [88,89]. PBA-based polymers possess high stability and can be structurally very versatile, thus they have been used as a glucose-sensing module for insulin delivery [90]. To achieve glucose-responsive insulin delivery, PBA is often incorporated in cross-linked polymeric matrices. These systems typically swell when negatively charged after the interactions with glucose [91]. Specifically, PBA-based polymers can bind glucose reversibly and this binding drives the swelling of polymeric matrices in particles containing insulin, thus improving insulin release. Self-assembled poly(ethylene glycol)-b-poly(acrylic acid-co-acrylamidophenylboronic acid) (PEG-b-(PAA-co-PAAPBA) micelles, reported by Wang et al., accelerate the release of insulin at higher glucose levels due to their swelling and disaggregating properties [92]. Ma et al. synthesized glucose-responsive micelles using a PBA-contained block copolymer PEG-b-P(AA-coAPBA) to complex with a PAA-based glycopolymer poly(acrylic acid-co-acrylglucosamine) (P(AA-co-AGA)). Such complex micelles with a P(AA-co-APBA)/P(AA-co-AGA) core and a PEG shell, showed fast glucose responsiveness and the glucose-sensitivity increased with a decrease in the amount of P(AA-co-AGA) in the compositions [93]. Kim et al. also developed a block copolymer containing polyboroxole as a saccharide-responsive polymeric domain that is able to self-assemble into a variety of nanostructures, including spherical and cylindrical micelles and polymer vesicles (polymersomes). In particular, they demonstrated that polymerosomes of this block copolymer can act as monosaccharide-responsive vehicles for insulin that disassemble and release insulin only in the presence of sugars [94]. Zhao et al. developed a glucose-responsive mesoporous silica nanoparticle (MSNP) to control precisely the double release of insulin and cyclic adenosine monophosphate (cAMP). cAMPs gluconic acid-modified insulin (G-Ins) proteins were immobilized on the outer surface of phenylboronic acid-functionalized MSNPs via reversible covalent bonding. In addition, G-Ins works as cap to encapsulate cAMP molecules inside the mesoporous channels. The innovative performance of this system relies on the fact that phenylboronic acid forms a much more stable cyclic ester with the adjacent diols of saccharides than with acyclic diols, and the presence of saccharides like glucose would trigger the release of both G-Ins and cAMP from particles [95]. 

Currently, Zenomics Inc. (North Carolina, NC, USA) is developing a transdermal patch technology, bearing microneedles loaded with insulin and a non-degradable glucose-responsive polymeric matrix, and fabricated via in situ photopolymerization. Under hyperglycemic conditions, phenylboronic acid units, within the polymeric matrix, reversibly form glucose–boronate complexes that induce the swelling of the matrix and weaken the electrostatic interactions between the negatively charged insulin and the polymer chains, promoting a rapid and appropriate insulin release [96]. 

## 5. Platforms to Accommodate or Protect Cell-Based Therapies for Insulin Release

Alternative to pharmacological treatment, ‘*β*-cell replacement therapy’ represents a more biological option to treat DM. This strategy involves transplantation of either the whole pancreas or solely islets. Since the first attempts in 1966 at pancreas transplantation and in 1974 at islets transplantation, *β*-cell replacement therapy has evolved to become a highly efficient procedure to restore endogenous and automated insulin secretion and protect against hypoglycemia, more effectively than any pharmacological therapy [97]. 

About 50,000 pancreas transplants have currently been performed worldwide and, according to the Scientific Registry of Transplant Recipients, the 1-year rate of pancreas graft survival is 90% when a pancreas and kidney are transplanted together, 83% when the pancreas is transplanted after the kidney, and 80% when the pancreas is transplanted alone [98,99]. Whole organ pancreas transplantation is invasive requiring a laparotomy, major vascular exposure and anastomosis, and the management of exocrine secretions from the pancreas via duodenoenterostomy. Other surgical complications associated with this procedure include infections, pancreatitis, and bleeding. All transplant patients require immunosuppression induction (e.g., monoclonal or polyclonal T-cell-depleting antibody therapy) and maintenance of immunosuppression with cyclosporine or tacrolimus, antimetabolites (mycophenolic acid), with or without corticosteroid administration [97,99,100]. 

In comparison to solid organ transplantation, islet transplantation is quicker, technically easier and less invasive. This strategy replaces only the islets (i.e., an endocrine component of the pancreas), which accounts only for 1%–5% of the total pancreatic mass. Since the introduction of the ‘Edmonton Protocol’ in 2000 [97,101], the results of islet transplantation have substantially and constantly improved, and today more than 1500 patients have received an islet transplant in 40 international centers. Approximately 70% of these patients no longer require daily insulin at 1 year after transplantation and the graft function was well maintained with 82% graft survival at 5 years [102,103]. 

Three different types of transplantation have been classified based on the different cell sources: auto-transplantation, allo-transplantation, and xeno-transplantation. (i) Auto-transplantation consists of transplanting a person’s own islets following a total pancreatectomy (i.e., for chronic pancreatitis [104], pancreatic arteriovenous malformation [105] and trauma [106,107]; (ii) allo-transplantation is performed when a diabetic patient receives islets from another individual(s) [108]; (iii) xeno-transplantation when a diabetic patient is transplanted with islets derived from an animal (i.e., pig) [109,110].

The clinically preferred site for the implantation of islets is the liver, where the portal vein is accessed with a catheter using a minimally invasive procedure and the islets infused into the liver [97]. Unfortunately, islet transplantation often requires 2 or more donor pancreases to provide a sufficient number of isolated islets to achieve insulin independence because, out of approximately 1 million islets in an adult human pancreas, only a half, or less, are successfully isolated and prepared for transplantation. Furthermore, similar to whole organ pancreas transplantation, islet transplantation requires the use of immunosuppressive drugs, and islet isolation and purification need to be performed under Good Manufacturing Practices conditions in an appropriate laboratory dedicated to supporting this procedure [99]. 

In 2015, a Phase III multicenter trial of the NIH CIT Consortium has been completed and confirmed that islet transplantation is a safe and effective treatment for patients with T1DM, although it was complicated by hypoglycemic unawareness or severe hypoglycemic events [111]. The FDA is evaluating the approval of this Biological Licensure Application that will be important for the future use of islet transplantation in the USA. However, the clinical application of islet transplantation has been limited by the high islet demand per patient, the need for immunosuppression, the high percentage of islet loss following transplantation and the overall long-term success. The introduction of foreign cells, which can be stem cells, that are able to produce insulin can lead to foreign body response, which generate an innate immune response that ultimately lead to the rejection of the transplant. In order to overcome these drawbacks, several efforts in tissue engineering have been taken to isolate and protect transplanted cells from the immune response while favoring optimal diffusion of oxygen, glucose, insulin, and other fundamental nutrients.

Tissue engineering is an interdisciplinary field, which applies the principles of engineering and life sciences toward the development of biological substitutes that restore, replace, maintain, or enhance tissue and organ functions [112]. The idea of tissue engineering for cell-based treatments for diabetes is to combine cells, specifically islets and/or *β*-cells, with biomaterials that provide a platform for the modulation and the control of the transplant site to create intimacy between the islets and the surrounding environment. These platforms include scaffolds which can mechanically support islets or cell encapsulation system which can protect islets from an immune-mediated attack (Figure 5). For example, polymeric bioscaffolds are porous and are designed to provide a solid support for islets that would allow cellular infiltration and formation of a vascular network within the transplant graft, but they are not intended to serve as an immune barrier. On the other hand, encapsulation systems have been shown to be able to prevent immune rejection of transplanted islets by precluding the ingrowth of blood vessels and islet revascularization which is important for long-term islet function [113,114]. 

### 5.1. Three-Dimensional (3D) Porous Bioscaffolds

Three-dimensional (3D) porous bioscaffolds have been developed to encourage islet engraftment and survival post-transplantation. Specifically, they provide the support that enables transplantation at extrahepatic and extravascular sites, which thus avoids the hypoxic environment of the liver and the instant blood-mediated inflammatory reaction (IBMIR) [115]. Several alternative sites such as kidney capsules [116], subcutaneous area, skeletal muscle [117], omental fat or gastric and intestinal submucosa have been studied, many of which show good physiological insulin and glucose responsiveness [118,119]. Bioscaffolds have been synthesized with a modular rigidity to resist collapse while maintaining an interconnected pore structure. While the large pores can accommodate islets (which are 150–400 μm in diameter) [118,120,121], the micropores can facilitate vascular ingrowth, helping islets to develop their own blood network [122]. The high surface area/volume ratio of the bioscaffold also favors cell integration within the host tissue and supports oxygen and nutrient diffusion from the surrounding environment.

Most of the materials used for building 3D bioscaffolds are either synthetic, such as PGA, PLGA or polydimethylsiloxane (PDMS), or natural, such as polysaccharides (i.e., chitosan, alginate, hyaluronic acid), inorganic polymers (i.e., hydroxyapatite), and natural proteins (i.e., collagen, fibrin, silk) and their configurations have ranged from beads to rods, to disks. For example, in preclinical studies, PLGA disks have been coated with islets in a matrigel suspension and transplanted into the epididymal fat pad of diabetic mice which resulted in these mice achieving normoglycemia compared with when only free islets or islets in a matrigel suspension were tested [123]. Similarly, Blomeier and coworkers reported that PLGA scaffolds with interconnected pores from 250 to 400 µm not only served as a platform for islet transplantation where they improved the function of extrahepatically transplanted islets compared to islets transplanted without a scaffold, but these scaffolds were also used to deliver bioactive molecules, to modify the microenvironment surrounding the transplanted islets to enhance islet survival and function [120]. Similarly, Jiang et al. demonstrated that the combined delivery of dexamethasone and islets in PDMS 3D bioscaffolds results in improved control of BGLs during the early transplant period [124]. As previously mentioned, the use of mechanical supportive materials is important to facilitate the development of a vascular network to supply the necessary nutrients to the transplanted islets. Indeed, Brady et al. evaluated the effect of a proangiogenic hydrogel-loaded macroporous PDMS scaffold containing islets that were implanted into the epididymal fat pad. The combination of the fibrin-based proangiogenic hydrogel and PDMS scaffold resulted in vessel branching and mature intra islet vasculature resulting in a quicker return to a normoglycemia state in animals [125].

Several studies have also shown that the extracellular matrix (ECM) serves as a key determinant in islet differentiation via the presentation of matrix-bound signals and the regulated delivery of soluble factors. Hence, studies are underway to elaborate and refine the design of the microenvironment in which transplanted islets are exposed for providing signals that help to promote islet engraftment (e.g., replacement and presentation of extracellular matrix proteins that are removed during islet isolation). For example, PLG bioscaffolds coated with ECM-proteins (i.e. collagen IV, laminin or fibronectin) have been shown to increase islet viability, decrease islet cell apoptosis, and enhance islet metabolism and glucose stimulation. Particularly, PLG scaffold coated with collagen IV, which is the principal component of the basement membrane, has been shown to enhance islet function and mediate early restoration of normoglycemia post-transplantation [126,127]. 

Finally, research is underway to evaluate the use of islet-like cells (derived from stem cells) with 3D microporous bioscaffolds. For example, Mao et al. have studied the subcutaneous transplantation of scaffolds seeded with the islet-like cells derived from the embryonic stem and have shown that these could improve 6 h fasted blood glucose levels and diabetic phenotypes in streptozotocin-induced diabetic SCID mice [128].

### 5.2. Encapsulation Devices

An islet-encapsulation device is based on the isolation and containment of one or several islets inside a hydrogel capsule, which is about several hundred micrometers in diameter. Specifically, these are classified into two types: macrocapsules and microcapsules. Macrocapsules are also divided in two types: intravascular and extravascular types. An intravascular macrocapsule is a perfusion chamber that is directly connected to the host artery and vein, while an extravascular one is a diffusion chamber containing a large number of islets [129]. Macrocapsules can be built in different shapes, including rod [130], tube [131] or sheet [132], and are easy to implant and remove with minimal risk. However, the permeability of macrocapsules is less than that of microcapsules given their thicker membrane even though they can accommodate multiple islets [133]. Microcapsules are thinner and show a better diffusion of oxygen and nutrient, resulting in better oxygen and nutrient transportation, but they are more difficult to remove completely when the device is infected. Ideally, encapsulation should be capable of protecting islets from the body’s immune responses (T cells, B cells or macrophages, antibody, and complement), while concurrently allowing the diffusion of small molecules like glucose, oxygen, and nutrients to islets, and insulin and waste products away from islets. 

The function of encapsulated islets depends on the materials selected. For example, alginate is probably the most widely used material for islet encapsulation. In 1980, Lim and Sun first demonstrated that islets microencapsulated in alginate were able to successfully achieve normoglycemia in diabetic rats for two-to-three weeks [134]. Since then, alginate microcapsules have been widely used for islet encapsulation because of its relative ease of gelling in the presence of divalent cations (typically Ca^2+^ or Ba^2+^). Alginate essentially incorporates islets in a semipermeable membrane and protects them from immune attack due to its negligible selectivity towards immune cells and other immunological factors, such as antibodies, that can potentially destroy islets [135]. Different strategies are used to modify alginate microcapsules, either with ECM molecules (like collagen) which can facilitate cell adhesion, or with cross-linkers (like PEG or poly-L-lysine; PLL) which can increase the stability of alginate microbeads. Indeed, PEG and PLL have been used to improve the alginate capsule by reducing plasma absorption and making a semi-permeable membrane. Alginate-functionalized with azide groups (N3-alginate) and HMW PEG terminated (MTD-PEG) were used to generate covalently linked alginate-PEG microbeads, resulting in non-toxic and more stable capsules compared to standard alginate beads [136]. In 1985, Goosen et al. developed a three-layer capsule consisting of alginate/PLL/alginate layers. Intraperitoneal transplantation of encapsulated islets reversed the diabetic state for at least one year compared to a single injection of unencapsulated islets, which was effective for less than two weeks [137]. 

Polysulphone (PSU) has also been used for islet encapsulation. PSU forms stable and highly porous polymers and has been used for renal dialysis [133]. Because of its hydrophobicity, its native form cannot be used for islet encapsulation and must be chemically modified. Lambert et al. showed that islets encapsulated in hydroxy-methylated PSU capillaries can release vascular endothelial growth factor (VEGF), which appears to be an important signal for the vascularization of the capillary material. Moreover, the roughness and porosity of the outer surface of PSU capillaries provide a site suitable for vascular tissue formation, which improves islet function post-transplantation [138].

Polyvinyl alcohol (PVA), a water-soluble synthetic polymer-forming stable and crystallizing hydrogel at low temperatures was also used by Inoue et al. in 1992 [139]. They showed that PVA membranes prevent immune rejection of pancreatic islets, favoring free diffusion of insulin, glucose, and nutrients [139,140]. Miyamoto et al. showed that the transplantation of a PVA capsule encapsulating porcine islets achieved normoglycemia in diabetic rats for 2 weeks [141] and modifying the device with an angiogenesis factor (i.e., fibroblast growth factor-2) helped with neovascularization around the capsule [142]. On the other hand, Qi et al. demonstrated that rat islets microencapsulated in a PVA device and xeno-transplanted into the peritoneal cavity of diabetic C57BL/6 mice improved the viability and function of islets restoring normoglycemia in transplanted mice up to 30 days post-transplantation [143]. 

The first transplantation with encapsulated human islets was performed by the University of California, Los Angeles (UCLA) group in 1999, in which cadaveric human islets were encapsulated in alginate microcapsules, and placed in the portal vein of a 38-year-old diabetic male. This initial success demonstrated that encapsulated islets were able to achieve glycemic control in T1DM patients [144]. In 2006, Calafiore et al. performed human islet microencapsulation using alginate and transplanted them into 10 T1DM patients in Italy. They used a specific protocol currently approved by the Italian Ministry of Health, which consist in a minimally invasive transplantation method by intraportal injection, under ultrasound imaging with local anesthesia, of clinical-grade sodium alginate without pyrogen and endotoxin and containing human islets with high purity and viability (over 80%) [145]. Then, the following study done in four T1DM patients transplanted with islets microencapsulated in barium alginate without immunosuppressant showed the same efficacy of non-encapsulated islets transplanted with immunosuppressant [146]. Despite advances in encapsulation and bioscaffold technology, these developments have not yet been meaningfully translated into clinical islet transplantation. Various factors are to blame: (a) graft hypoxia, (b) host inflammatory response, (c) fibrosis, (d) inappropriate choice of the biomaterial type, (e) lack of standard guidelines and (f) failure of the post-transplantation device. To date, active investigations are underway, such as the use of porcine islets [147,148], stem cells [149], development of revascularized implants [150,151], islet nanocoating [152], and multilayer encapsulation [153], to improve the existing technology and achieve the goal of a permanent cure.

## 6. Future Directions and Conclusions

Recently, one of the most innovative and alternative strategies for the treatment of diabetes patients addressing the glucose-responsive system combine nano-, micro-technology and external remote triggers (i.e., pulsed focus ultrasound (pFUS) or radio waves). 

Ultrasound is regularly used in medical applications for diagnostic imaging techniques and has been shown to facilitate controlled drug delivery, providing both long-term sustained and fast on-demand responses for triggering the release of encapsulated drugs and to enhance and promote transdermal drug delivery [154,155]. The variation of release kinetics can mostly be attributed to ultrasound-induced heating, acoustic streaming, or mechanical effects, such as nucleation, growth, and collapse of gas bubbles, a process referred to as inertial cavitation [155]. Di et al. have developed a spatiotemporally-controlled insulin delivery system consisting of injectable polymeric nano-network, which can be effectively triggered to release insulin upon focus ultrasound (FUS)-mediated administration. This system provides an extremely useful tool for non-invasive, rapid and pulsatile regulation of blood glucose levels for diabetes treatment. Using biocompatible and biodegradable PLGA as a matrix material, Di et al. have demonstrated that the insulin encapsulated in the nano-network formulation, with 30-sec FUS-mediated activation, can effectively regulate blood glucose levels of type 1 diabetic mice in a long-term and pulsatile delivery manner [156,157]. 

Alternatively, Stainley et al. investigated a non-invasive and non-pharmacological approach based on radio waves. They decorated a temperature-sensitive channel, TRPV1, with an antibody-coated with iron oxide nanoparticles, which are heated in a low-frequency magnetic field. When iron-nanoparticles were exposed to radio waves, the local temperature increase, opening temperature-sensitive TRPV1 channels, driven by a calcium-sensitive promoter [158].

In view of the above, nano- and micro-technologies are helpful in improving the therapeutic efficacy of insulin, achieving a self-regulated drug-release profile as a function of glucose concentrations because of glucose-sensing molecules, and increasing drug physicochemical and biological stabilities. On the other hand, tissue engineering is paving the way to new strategies that could encourage the reconstruction of complex tissue architecture, make possible long-term *β*-cell replacement, and limit the need for immunosuppressive drugs. Despite these findings, many of the strategies described and currently in use lack a definitive efficacy, and the insulin release based on glucose-demand may fail, resulting in hypoglycemia. Thus, the best way to address future challenge is by taking a multidisciplinary approach, which combines nano- and micro-technologies with tissue engineering strategies to develop devices, enabling an artificial pancreas-like system to be obtained.

## Figures and Tables

**Figure 1 nanomaterials-10-00789-f001:**
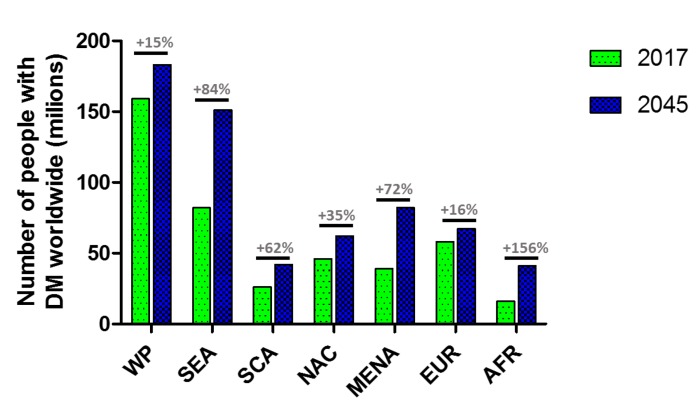
Number of people (millions) aged 20–79 years old with diabetes mellitus DM estimated by IDF, 2017 and 2045. (IDF = International Diabetes Federation, AFR = Africa, EUR = Europe, MENA = Middle East and North America, NAC = North America and Caribbean, SCA = South and Central America, SEA = South East Asia, WP = Western Pacific) [1].

**Figure 2 nanomaterials-10-00789-f002:**
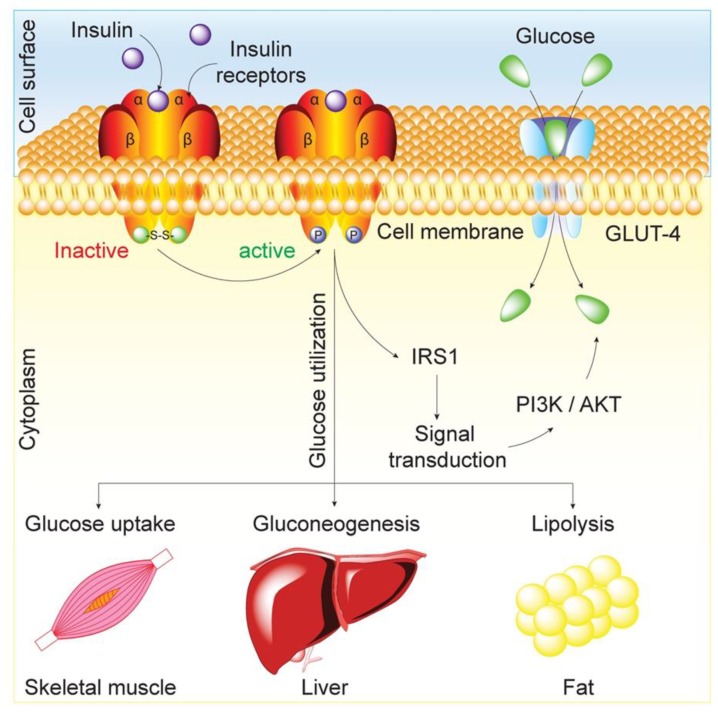
Physiological mechanism of insulin. The insulin receptor is a tyrosine kinase receptor which is a heterotetrameric glycoprotein, consisting of 2 extracellular *α* and 2 transmembrane *β* subunits linked together by disulfide bonds. The α subunits carry insulin binding sites, while *β* subunits have tyrosine kinase activity, involved in intracellular signaling. When insulin binds the *α* subunits of its receptor, the PIK3K-signaling pathway is activated. It stimulates glucose transport across cell membrane by translocation of glucose transporter GLUT4 to the plasma membrane, increasing rate of glucose influx, promoting glycogen synthesis and lipogenesis.

**Figure 3 nanomaterials-10-00789-f003:**
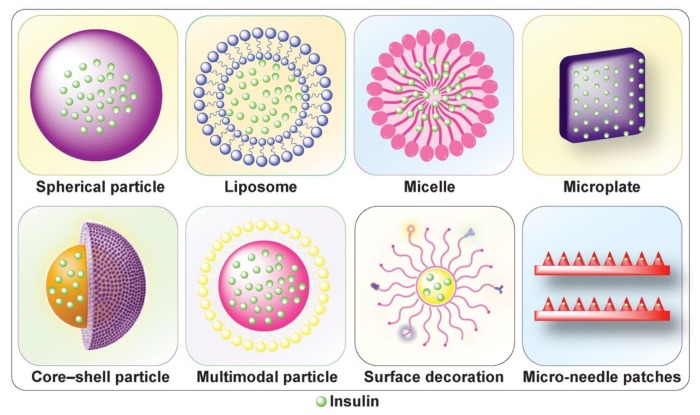
Schematic illustration of various types of macro- or nano-delivery platforms.

**Figure 4 nanomaterials-10-00789-f004:**
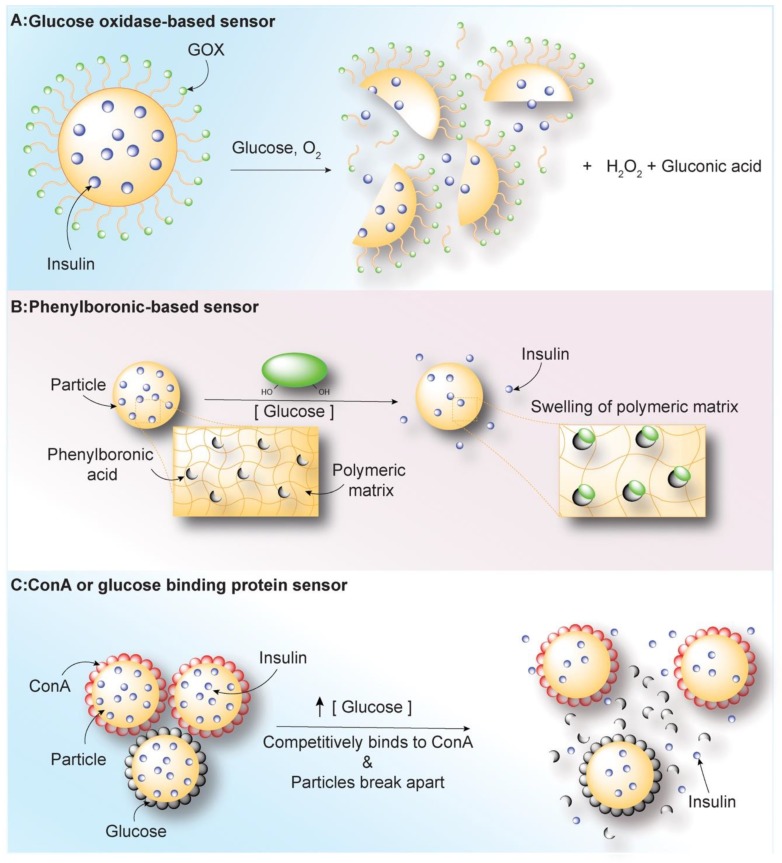
Particle-based glucose sensors. (**A**) Glucose oxidase (GOx)-based systems show high affinity for glucose, GOx catalyzes the oxidation of glucose to gluconolactone firstly with the production of toxic intermediate hydrogen peroxide (H_2_O_2_), and then gluconolactone is rapidly hydrolyzed to gluconic acid in an aqueous environment. The local acidification in response to elevated glucose levels trigger the release of insulin from particles. (**B**) Phenylboronic acids are Lewis acids that can bind reversibly to cis-1,2 or cis-1,3 diols. This binding drives the swelling of polymeric matrix of particles containing insulin, improving insulin release. (**C**) Concanavilin-A (ConA) is a natural protein containing four binding sites for sugar. Particles containing ConA can release insulin via a competitive binding between glucose and Conc-A upon glucose exposure.

**Figure 5 nanomaterials-10-00789-f005:**
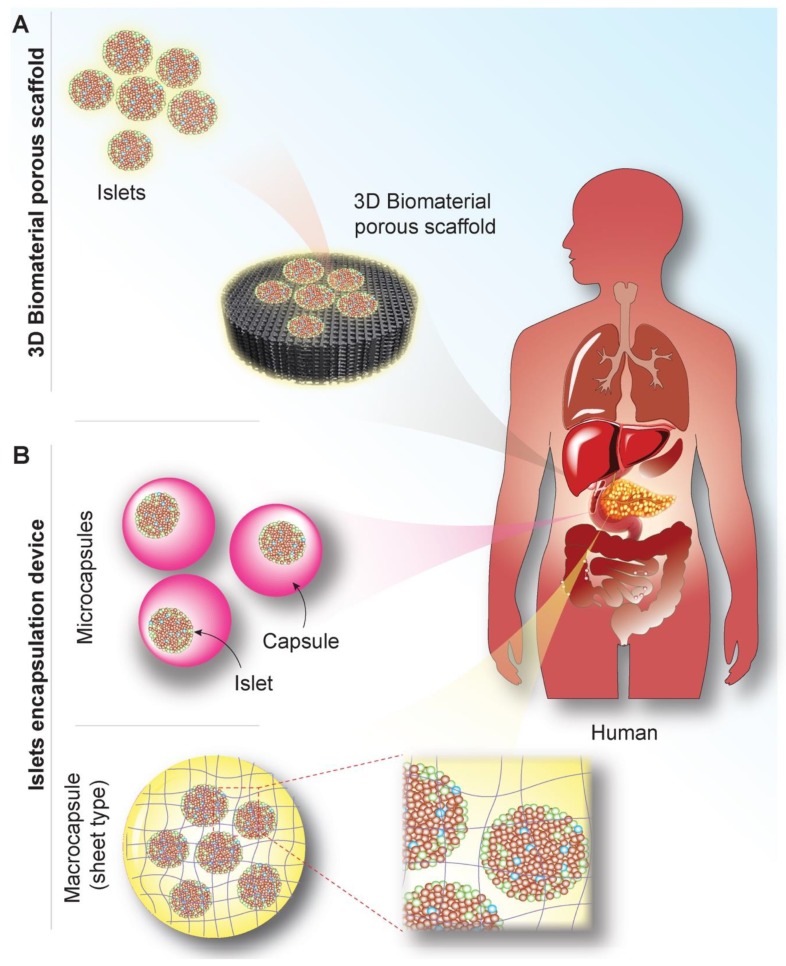
Schematic illustration of different approaches investigated for islet-based tissue engineering. (**A**) Three-dimensional (3D) porous polymeric bioscaffold designed to provide mechanical support for islets enabling transplantation at extrahepatic and extravascular sites. (**B**) Encapsulation systems, including micro- and microcapsules, able to prevent immune rejection of transplanted islets.

**Table 1 nanomaterials-10-00789-t001:** Insulin variants.

Insulin Variants	Onset of Action	Peak of Action	Duration of Action	References
**Rapid-acting Insulin** *Lispro* *Aspart*	5–15 min	1–2 h	4–5 h	[10]
**Short-acting Insulin** *Regular*	30–60 min	2–5 h	5–8 h	[8]
**Intermediate-acting Insulin** *NPH*	1–2 h	4–8 h	10–20 h	[11]
**Long-acting Insulin** *Detemir* *Glargine*	1–2h	Relatively flat	14–24 h	[11]

**Table 2 nanomaterials-10-00789-t002:** Platform for carrying insulin: advantages and disadvantages.

Platforms for Carrying Insulin	Advantages	Disadvantages	References
**Nano-scale carrier**	Enhance drug stability (i.e., preventing insulin enzymatic and hydrolytic degradation) Enhance permeation of drug in tissue and bloodstream Extend drug circulation Susceptible to cell-uptake Sustain insulin release Suitable for systemic and oral administration	Short-term insulin release Not suitable for tissue implantation	[15,20,21,22,23,24,25,26,27,28,29,30]
**Micro-scale carrier**	Enhance drug stability (i.e., preventing insulin enzymatic and hydrolytic degradation) Sustain and long-term release of insulin Suitable for oral and nasal administration and optimal for tissue implantation Ability to entrap smaller particles (i.e., hierarchical system) Long-term retention in the implantation site	Not suitable for systemic administration Not susceptible to cell-uptake	[33,34,35,36,37,38,39,40,41,42,43,44,45,46,47,48,50,51,52,53]
**Microneedle**	Able to circumvent subcutaneous barrier Suitable for transdermal delivery of insulin Improve drug penetration through the skin Avoid first-pass metabolism Painless and more rapid injection site healing compared with conventional hypodermic needles	Less precise dosage accuracy compared to hypodermic needles Drug-delivery issues associated with non-vertical application of the microneedle to the skin Potential breakage of microneedle tip within the skin	[54,55,56,57,58,59,60,61,62]

**Table 3 nanomaterials-10-00789-t003:** Glucose-sensing molecules: advantages and disadvantages.

Glucose-Sensing Molecules	Advantages	Disadvantages	References
**Glucose** **oxidase-based sensor**	High glucose specificity	Slow response rates Susceptible to oxygen and pH fluctuations	[72,79]
**Phenylboronic-based sensor**	Structurally very versatile Rapid response rates High stability	Lack of glucose specificity	[90,91,92]
**ConA or glucose binding protein sensor**	High glucose specificity Rapid response rates	Instability Host immune response High cost	[83,84,87]

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
