# Peer review of "Emerging Nano- and Micro-Technologies Used in the Treatment of Type-1 Diabetes"

_nanomaterials, 2020, doi:10.3390/nano10040789_

Round 1
Reviewer 1 Report
The manuscript ", written by Primavera et al., describes the utilization of nano/micro materials and various scaffolds in a treatment of diabetes, Type-1. The manuscript is very well written and carefully prepared. I have only few minor comments:
- Data in figures should be adequately cited (e.g. figure 1). I thus suggest to add references also to figure captions.
- Authors should also mention if the described methods ale already implemented / at least partially/ in clinical studies or if they are on the market.
Author Response
Please see the attachment in the box.

Reviewer 2 Report
Comment to author
In this manuscript, Primavera and coworkers reviewed recent progress on Type-1 diabetes and discussed how nano- and micro-technologies, as well as tissue engineering approaches, can overcome many of these challenges and help contribute to an artificial pancreas-like system. Although the author summarizes a wide range of approaches, the information is often confusing. The authors should try to convey to the reader in an easy-to-understand manner, such as summarizing each method using diagrams and comparing them in tables.
1) The various types of macro- or nano-delivery platforms were summarized in Figure 3.  Authors need to specify the advantages and disadvantages of each platform in the new table.
2) The modulate insulin delivery section is well organized. This section should likewise summarize the advantages and disadvantages of each approach in the new table.
3) Overall, for each method, the authors have successfully compiled previous works. However, it is necessary to describe the chemical mechanism or detection limit of each method.
4) The author needs to describe the information of the equipment necessary for performing each detection method and delivery method. Authors must also provide information such as whether the platforms for carrying insulin are commercially available.
Author Response
Please see the attachment in the box.
